# Transfer Learning for Radio Frequency Machine Learning: A Taxonomy and Survey

**DOI:** 10.3390/s22041416

**Published:** 2022-02-12

**Authors:** Lauren J. Wong, Alan J. Michaels

**Affiliations:** 1Hume Center for National Security and Technology, Virginia Tech, Blacksburg, VA 24061, USA; ajm@vt.edu; 2Intel AI Lab, Santa Clara, CA 95054, USA

**Keywords:** machine learning (ML), deep learning (DL), transfer learning (TL), radio frequency machine learning (RFML)

## Abstract

Transfer learning is a pervasive technology in computer vision and natural language processing fields, yielding exponential performance improvements by leveraging prior knowledge gained from data with different distributions. However, while recent works seek to mature machine learning and deep learning techniques in applications related to wireless communications, a field loosely termed radio frequency machine learning, few have demonstrated the use of transfer learning techniques for yielding performance gains, improved generalization, or to address concerns of training data costs. With modifications to existing transfer learning taxonomies constructed to support transfer learning in other modalities, this paper presents a tailored taxonomy for radio frequency applications, yielding a consistent framework that can be used to compare and contrast existing and future works. This work offers such a taxonomy, discusses the small body of existing works in transfer learning for radio frequency machine learning, and outlines directions where future research is needed to mature the field.

## 1. Introduction

The past several years has observed a rise in the application of machine learning (ML) and deep learning (DL) techniques to the wireless communications domain for applications including spectrum awareness, cognitive radio, and networking. Such techniques, loosely defined radio frequency machine learning (RFML) [1], provide increased performance and flexibility while reducing pre-processing and pre-defined expert features when compared to traditional signal processing techniques [2,3,4]. The vast majority of these RFML works train ML/DL models from random initialization, and use supervised learning techniques. Such an approach assumes the availability of a large corpus of labeled training data, which is representative of anticipated observations once deployed, including radio frequency (RF) hardware variations and channel effects. When this assumption breaks down, existing works have shown limited ability to generalize to new hardware effects and channel conditions. That is, if the hardware effects and/or channel conditions stray outside of those represented in the training data, performance drops severely. For example, preliminary results given in [5] showed that the performance of convolutional neural network (CNN) and long short-term memory (LSTM)-based automatic modulation
classification (AMC) algorithms trained on data from a single transmitter/receiver pair not only varied significantly across different transmitter/receiver pairs but also dropped by as much as 8% when tested on augmented data captured from other transmitter/receiver pairs. Furthermore, results in [6] showed that a CNN-based specific emitter identification (SEI) model learned to correlate channel distortions with emitters, rather than learning the characteristics of the emitters themselves, and consequently performed poorly under different channel conditions.

Transfer learning (TL) aims to overcome these obstacles by utilizing prior knowledge gained from a *source* domain and task to improve performance of a model for a “similar” *target* domain and task, when compared to training only on the target domain and task from random initialization (Figure 1). Although the term “similar” is ill-defined, the use of TL techniques has resulted in exponential growth in fields such as computer vision (CV) and natural language processing (NLP), demonstrating a wealth of performance gains that have yet to be fully utilized in RFML [7].

While TL methods are beneficial in a wide variety of learning scenarios, TL shines when sufficient training data are not available in the target domain, yet similar source data are available from which knowledge of the target domain/task can be gleaned. Therefore, TL provides an avenue for increased performance with reduced captured training data and for using trained models across a wider variety of hardware platforms and channel conditions without retraining. For example, large captured training datasets have been shown to yield the greatest performance, but require several orders of magnitude more time to create when compared to synthetic and augmented datasets [8]. TL may enable comparable performance with less captured training data by using prior knowledge gained from synthetic, augmented, or other captured training datasets. Furthermore, unlike in the fields of CV or NLP, TL is likely a requisite technology for realizing online and distributed RFML algorithms, as behavior learned on one platform will be distinctly impacted by RF hardware and will, therefore, vary from platform to platform.

While several TL taxonomies and surveys can be found in the literature which examine TL more generally [9,10] and in the context of specific modalities or learning paradigms such as CV [11,12], NLP [7], and reinforcement learning (RL) [13], no studies to our knowledge have examined TL in the context of RFML. In contrast, this work offers a domain-specific TL taxonomy for RFML that provides a framework for comparing and contrasting existing works and underscores unique facets of RFML, which may warrant deliberate TL algorithm design and yield significant performance gains. In addition, this work surveys the small body of existing works and suggests directions for future research.

This paper is organized as follows: Section 2 provides the requisite notation and definitions used in the remainder of the work, background for why TL is necessary to accelerate research in RFML, and examples of how the terms domain and task can be interpreted in an RFML context. In Section 3, we briefly discuss the general TL from which our RFML-specific taxonomy is derived. Next, Section 4 presents RFML-specific TL taxonomy, discusses specific motivations for RFML TL applications, and highlights the adaptations made to the general taxonomy introduced previously. Moreover, in this section, we survey the small body of existing works in TL for RFML, discuss the methods used to achieve transfer in these works, and highlight initial trends observed across the works. Section 5 outlines a few key suggested directions for future research in TL for RFML outside of algorithmic development. Finally, Section 6 concludes the study.

## 2. Definitions

The term RFML has been used in the literature to describe any application of ML to the RF domain, including cognitive radio applications and approaches relying on classical signal processing techniques and expert-defined feature extraction [14]. However, in this work, we narrow the scope of works discussed to align with the first definition of RFML in [1] and those used in [14]. More specifically, we define RFML to be approaches, techniques, and works aimed at reducing the use of expert-defined features and the amount of prior knowledge needed for the intended RF application, and we primarily discuss DL-based works that use raw RF input. As discussed further in Section 4, TL has been used more in the context of cognitive radio and/or expert feature-based algorithms but has been used very little in the context of RFML, as defined here.

As commonly accepted in the literature, this work uses the TL notation introduced in [9]: A domain D={X,P(X)} comprises input data *X* and the marginal probability distribution over data P(X), where $X=\{x_{1},\ldots ,x_{n}\}\in X$ and where X denotes the input space. The task T={Y,P(Y|X)} comprises the label space *Y*, and conditional probability distribution P(Y|X) learned from the training data pairs $\left\{x_{i},y_{i}\right\}$ such that $x_{i}\in X$ and $y_{i}\in Y$. Generally, for RFML, the domain consists of RF hardware and channel environment, and the task comprises the application being addressed, including the range of possible outputs. Example elements of RFML domains and RFML tasks are provided in Table 1.

The source domain and task, denoted $D_{S}=\left\{X_{S},P\left(X_{S}\right)\right\}$ and $T_{S}=\left\{Y_{S},P\left(Y_{S}|X_{S}\right)\right\}$, are those defined during the initial training process. That is, the source domain and task describe initial training data and labels. The target domain and task are denoted $D_{T}=\left\{X_{T},P\left(X_{T}\right)\right\}$ and $T_{T}=\left\{Y_{T},P\left(Y_{T}|X_{T}\right)\right\}$, and they describe the intended use-case of the trained ML model. Note that labeled data may or may not be available for the target domain and task or may only be available in limited quantities.

Traditional supervised ML techniques assume that $D_{S}=D_{T}$ and $T_{S}=T_{T}$ [9], allowing *direct transfer* to be employed with success. That is, we can use the model trained for the source domain and task for the target domain and task with no modification. However, in the context of RFML, inherent hardware variations and channel effects all but guarantee that $D_{S}\neq D_{T}$, unlike in the fields of CV and NLP. TL is motivated by this mismatch between the source and target domains and/or tasks, inhibiting direct transfer. More specifically, the aim of *TL* is to leverage the knowledge $P(Y_{S}|X_{S})$ obtained using $D_{S}$ and $T_{S}$ to improve the performance of $P(Y_{T}|X_{T})$ on $D_{T}$ and $T_{T}$ [9].

TL is feasible because a model trained on a source domain and task has learned generic knowledge about the structure of raw RF signals through the source domain/task, which may be used as prior knowledge to solve the target task. The utility of these previously learned features for solving the target task is dependent on the “similarity” between the source and target tasks and domains. On the one hand, the learned features used to perform AMC on a low cost IoT transceiver are likely quite similar to those used to perform AMC on a high cost military-grade transceiver. On the other hand, the learned features used to perform AMC are likely different from those learned when performing SEI. As mentioned previously, the “similarity” between the source and target domains/tasks is not well-defined, but can be thought of as a continuous two-dimensional spectrum, depicted in Figure 2, and dictates the success of TL.

To frame this discussion, consider the following four scenarios and examples in which $D_{S}\neq D_{T}$ and/or $T_{S}\neq T_{T}$ can occur:$P\left(Y_{S}|X_{S}\right)\neq P\left(Y_{T}|X_{T}\right)$—the source and target tasks have different conditional probability distributions. This most commonly manifests in the form of unbalanced datasets, where a subset of classes has more examples in the source dataset than the target dataset or vice versa. A simple example might be transferring an AMC model between two datasets, both of which only contain BPSK and QPSK signal types. However, the source dataset contains 70% BPSK signals and 30% QPSK signals, while the target dataset contains 30% BPSK signals and 70% QPSK signals.$Y_{S}\neq Y_{T}$—the source and target tasks have different label spaces. For example, the target task contains an additional output class (i.e., for an AMC algorithm, and the source task is a binary BPSK/QPSK output set, while the target task includes a third noise-only class). Alternatively, the target task may be completely unrelated and disjoint from the source task (i.e., the target task is to perform SEI, while the source task was to perform AMC); therefore, the label spaces are also disjoint.$P\left(X_{S}\right)\neq P\left(X_{T}\right)$—the source and target domains have different data distributions. An example of such a scenario includes a transfer of models from one channel environment to another, as described further in Section 4.1.1.$X_{S}\neq X_{T}$—the source and target feature spaces differ. An example includes performing SEI using the same set of known emitters but using different modulation schemes in the source and target domain.

These scenarios are not mutually exclusive. That is, for any given TL setting, several of the above scenarios may be encountered. For example, when $Y_{S}\neq Y_{T}$ (Scenario 2) and/or $P\left(X_{S}\right)\neq P\left(X_{T}\right)$ (Scenario 3), the source and target tasks typically also have different conditional probability distributions (Scenario 1).

## 3. Related TL Taxonomies

Before presenting a TL taxonomy for RFML in the next section, we overview the general TL taxonomy presented in [9] from which our taxonomy builds. This taxonomy, or some adaptation thereof, is used in a number of fields including NLP [7] and CV [11]. While there are a number of ways to categorize TL problems, refs. [9,10] categorizes the broad field of TL into three sub-fields—*unsupervised, inductive, and transductive*—each characterized by the availability of training data in the source and/or target domains and whether or not the source and target tasks differ.

In *unsupervised TL*, no labeled data are available in either the source and target domains. The source and target tasks can be the same or different. *Inductive TL* settings are characterized by the availability of labeled data in the target domain, when the source and target tasks differ. Labeled data may or may not be available in the source domain. *Inductive TL* is further broken out into the following:*Self-taught* methods that address settings where no labeled data are available in the source domain;*Multitask learning* that assumes the availability of labeled data in both the source and target domains and in which the source and target tasks are learned simultaneously;*Sequential learning*, which also assumes the availability of labeled data in both the source and target domains; however, the source task/domain is learned first and the target task/domain is learned second.

It should be noted that sequential learning was not included in the taxonomy presented by [9] but was detailed in [7], as it is an oft-utilized approach in the DL literature and a critical component of the meta-learning, lifelong learning, and representation learning fields.

In *transductive TL* settings, no labeled data are available in the target domain, while the source and target tasks are the same. *Transductive TL* is further broken out into the following:*Domain adaptation*, under which the source and target domains differ;*Sample selection bias*, also known as *covariance shift*, which refers to when both source and target domains and tasks are the same, but the source and/or target training dataset may be incomplete or small.

## 4. An RFML-Specific Taxonomy

The proposed TL taxonomy for RFML is shown in Figure 3 and is adapted from the general taxonomy discussed previously to contain TL contexts most relevant to the current state-of-the-art RFML algorithms. More specifically, given the limited use of unsupervised and self-supervised algorithms in RFML literature, this taxonomy assumes the availability of some labeled data in both source and target domains, although the size of these labeled datasets may be limited. This restricts the discussion herein to inductive TL techniques. However, this taxonomy can easily be expanded to include transductive TL techniques, as needed.

In addition to limiting the discussion to inductive TL techniques, three key changes have been made specific to the RF domain: First, while domain adaptation is considered a transductive TL approach in the general taxonomy of [9], we consider inductive TL techniques for the purpose of domain adaptation. That is, we discuss domain adaptation approaches that make use of the labeled source as well as target data. Second, similarly to the RL TL taxonomy presented in [13], the area of domain adaptation is further divided into three categories: *environment adaptation*, *platform adaptation*, and *environment platform co-adaptation*. This alteration specifies the type of domain change and highlights that an environmental shift (i.e., a channel change) is vastly different than a change in transmitter or receiver hardware, as described further in the subsequent subsections. Third, similarly to [7], sequential learning is included in this taxonomy to provide a counterpart to multitask learning. For the sake of clarity, representative examples for each TL setting are described in Table 3 and are expounded upon in the following subsections with parallels drawn to other modalities where appropriate.

### 4.1. Domain Adaptation

When the source and target tasks are the same, but the source and target domains differ, we require *domain adaptation* in the form of *environment adaptation*, *platform adaptation*, or *environment platform co-adaptation*, and each are described in the following subsections. More specifically, *domain adaptation* techniques are needed when the label space remains constant (i.e., $Y_{S}=Y_{T}$) and the conditional probability distributions learned from the source and data sets is the same (i.e., $P\left(Y_{S}|X_{S}\right)=P\left(Y_{T}|X_{T}\right)$), but the source and target domains have different data distributions (i.e., $P\left(X_{S}\right)\neq P\left(X_{T}\right)$) and/or the source and target feature spaces differ (i.e., $X_{S}\neq X_{T}$). However, the cause of different source and target data distributions and/or feature spaces can be caused by either a change in platform (i.e., transmitter and/or receiver hardware) or environment.

#### 4.1.1. Environment Adaptation

In the context of RFML, the aim of *environment adaptation* is to adapt a learned model to a changing channel environment, while holding the transmitter/receiver pair(s) constant. Environmental factors such as time of day, temperature, atmospheric conditions, channel type, and any movement of the transmitter and/or receiver may potentially create variations in signal capture which has the potential to affect the learned behavior of an RFML system. Consider the representative example of moving a transmitter/receiver pair equipped with an AMC model from an empty field to a city center. Although the transmitter/receiver pair stays constant, we move from a line-of-sight, likely AWGN, channel to an environment with significant multipath effects and interference from neighboring devices. Such an example is similar to performing image classification indoors versus outdoors [15] or utilizing image classification algorithms in environments where the captured image may degraded by weather conditions [16].

Few works have examined the impact of a changing environment on RFML performance, and as a result, little is known about the extent to which the parameters given above may prevent transfer between environments. However, existing work used finetuning techniques to successfully transfer RFML models from one real environment to additional real environments [17]. More specifically, in [17], a robust DL-based spectrum sensing framework was proposed that used techniques similar to those used for *sequential learning* to adapt pretrained models to changing wireless conditions with little-to-no labeled target data.

If we consider the use of spectrograms as input to a DL model, rather than raw RF data, additional work presented in [18] used a CNN-based support vector machine (SVM) approach to perform non-cooperative spectrum sensing. In this study, an AlexNet inspired CNN was used as a naive feature extractor, and a linear SVM was used to determine whether or not the spectrum band-of-interest was occupied using the features extracted by CNN. When the environment or location changed, the initial layers of the CNN feature extractor were frozen, while the remaining layers and the SVM were retrained using data from the new environment or location. Results showed that TL reduced the number of spectrograms needed to achieve the same performance without TL, and this was most significantly observed when transferring from environments with low SNR levels to environments high SNR levels. Some performance improvements were also observed when transferring between environments with similar SNR levels. However, performance typically degraded when transferring from high SNR levels to low SNR levels, a phenomenon known as negative transfer [19,20].

#### 4.1.2. Platform Adaptation

In contrast to environment adaptation, the aim of *platform adaptation* is to overcome changes in transmitter/receiver hardware while holding the channel environment constant. Variations in hardware non-linearities, IQ imbalances, or frequency; phase; and/or timing offsets all have the potential to inhibit model transfer between platforms. Additionally, note that while the receiver hardware will always be user-controlled, the transmitter may or may not be. That is, if we are a third party listener, changes in transmitter hardware will be outside of our control, further complicating the task.

A representative example of platform adaptation includes transferring an AMC model between UAVs. Presuming, when UAVs are flying in the same vicinity, the channels they encounter will be similar. Additionally, the received signals on both platforms will be affected by Doppler shifts. However, small hardware variations caused by manufacturing inconsistencies, age, settings, etc., will cause variations between the signals received on each platform. While one might draw parallels between transferring models between RF platforms and between cameras capturing images for CV algorithms, it seems that, assuming that the two cameras are capable of capturing images at the same resolution, such a transfer does not affect performance in any significant manner [21]. However, Ref. [5] showed that directly transferring learned models between transmitter and receiver pairs diminished performance by as much as 7%, even when augmentations, such as adding noise, frequency offsets, and resampling, were applied to the training data. These results have been echoed in several subsequent works, both in the context of AMC [8] and SEI [22], which have aimed to mitigate performance degradation through data augmentation/transformation, data preprocessing, or training over a variety of platforms. However, despite the growing body of work recognizing the need for platform adaptation methods, little work has been performed to identify the impact of changing hardware platforms on RFML performance or to develop methods for transfer between hardware platforms, as discussed in further detail in Section 5.

#### 4.1.3. Environment Platform Co-Adaptation

Finally, *environment platform co-adaptation* combines the challenges of environment adaptation and platform adaptation with the goal of transferring a learned model to a new channel environment, as well as to a new transmitter/receiver pair(s). As a representative example, consider transferring an AMC model between an RFML-enabled ground station and a UAV. In such a scenario, not only will the change in hardware impact the received signals and resultant performance but the channel environments encountered by the two devices will also differ significantly. Due to the fact that changes in CV platform (i.e., cameras) do not impact performance in the same manner that changes to the RF platform do, as discussed above, *environment platform co-adaptation* is a scenario that is not typically discussed in the CV literature but is akin to techniques aimed at domain adaptation using drawings or clip art as a source domain and real images as the target domain [23]. Conceptually, this is similar to transferring models between synthetic and captured data in the RF space.

In the context of RFML, existing works in the area of environment platform co-adaptation have primarily taken the form of DL models pre-trained on synthetic data and finetuned using captured data. More specifically, Refs. [24,25] examine transferring residual and autoencoder (AE)-based models from synthetic to real environments for AMC and channel model estimation problems. These works finetune varying amounts of the pre-trained neural network (NN)—only the final layer in [24] and the latter half of the NN in [25]—for a small number of epochs with a small learning rate, taking cues from the CV literature [26]. If we categorize power spectrums as raw RF data, Ref. [27] has also examined the transfer of CNNs from synthetic to real environments for signal detection. In this work, the entire CNN is tuned again for a small number of epochs with a small learning rate.

Recent work has also confirmed the intuitive result that the order in which datasets are trained is critical to achieving successful transfer of learned behaviors [28]. More specifically, in [28], the authors showed that when comparing the performance of an AMC model trained on synthethic data, augmented data, captured data, or some combination thereof, the best performance on captured test sets (which are representative of what will be observed once deployed) is achieved when pre-training on synthetic datasets and using captured data for finetuning. That is, synthetic and augmented data are best for pre-training, while captured data, which is typically smaller in quantity anyhow, are best for finetuning. Such results do make the realistic assumption that the final trained model will be evaluated/tested/deployed on real captured data, the implications of which are discussed further as a part of sequential learning in Section 4.3.

### 4.2. Multitask Learning

The aim of *multitask learning* is to learn differing source and target tasks simultaneously and is typically characterized by the use of more than one loss term during training. This encourages the model to learn more general features that are useful in multiple settings. For example, an RFML model trained to simultaneously perform signal detection and AMC will likely learn more general features about signal structure and modulation than an RFML model trained to perform only one of these tasks.

Multitask learning has perhaps been explored more frequently than any other TL setting in the context of ML-enabled wireless communications using expert-defined features. For example, in [29], a *multitask learning* architecture is designed for joint jamming detection/localization and link scheduling; in [30,31], CNN and recurrent neural network (RNN) architectures are trained to perform Wi-Fi and cellular traffic forecasting, predicting the maximum, minimum, and average load and the load across neighboring cells, respectively, and work in [32] used CNNs to perform indoor Wi-Fi localization.

However, examples of multitask learning in the context of RFML, as defined in this work, are far fewer. The limited body of works include an approach for end-to-end communications presented in [33], as well as several that have explored multitask learning as a method to both improve the explainability and accuracy of models trained to perform automatic modulation classification (AMC). More specifically, in both [34,35], modulation classes are broken into subgroups, either by modulation type (i.e., linear, frequency, etc.) or in order to separate the modulation schemes that cause the most confusion (i.e., 16QAM and 64QAM); moreover, in [36], concept bottleneck models were used to provide inherent decision explanations while performing AMC via the prediction of a set of intermediate concepts defined prior to training.

### 4.3. Sequential Learning

Finally, *sequential learning* describes the setting in which a source task is learned first, and the aim is to transfer the pre-trained model to a different target task, typically via fine-tuning techniques [26], similarly to those used for *domain adaptation*. In ever-changing wireless conditions, sequential learning will be a critical component of future online, lifelong, and meta-learning techniques for RFML systems. For example, adding output class(es) to a pre-trained AMC model can be considered a representative example of sequential learning. Such an approach was examined in [37] and could be extended by performing a successive refinement of models by adding a single signal type to the task at a time. More specifically, in [37], sequential learning techniques were used to fine-tune a pre-trained residual CNN for 190 entirely new categories/output classes, with as few as 50 to 500 samples per category, versus a model trained from random initialization. (The pre-trained model had been trained for using 13,000,000 training samples from over 5000 categories.) Results showed that the fine-tuned models not only converged over an order of magnitude faster than the model trained from random initialization, but achieved higher test accuracies as well.

Additional work in [38] examined the use of sequential learning methods for adapting pre-trained SEI models for intended use cases, including tuning for changes in emitters (i.e., output classes) and protocols used. More specifically, this work built upon an existing architecture, RiftNet [39], using supervised pre-training and fine-tuning methods similar to those discussed above, as well as unsupervised pre-training and transfer learning methods, such as the use of reconstruction losses and manifold clustering, for novel device detection/classification. This work further examined the impact of source/target dataset size, the number of source/target output classes, and changes in protocol between the source and target on transferability. When using supervised pre-training and fine-tuning methods, the results showed that pre-training on larger, more diverse source datasets provided the best transfer learning result, hypothesizing that such models learned the most generalizeable features for the domain. The fine-tuning of these pre-trained models outperformed baseline classifiers trained from random initialization in most all cases, and performance was best when only the relevant output classes were retained and extraneous output classes were removed. As expected, the larger source and target datasets yielded higher performance, with the size of the source dataset having a slightly larger impact on end performance than the size of the target dataset. However, overfitting was common during the fine-tuning process, requiring care and attention in the setting of hyperparameters and use of early stopping and/or checkpoint methods. Additionally, transfer was more challenging between protocols, requiring additional fine-tuning steps and resulting in low top-1 accuracy. When using unsupervised reconstruction-based transfer learning methods, results showed that the use of multiburst processing, batching five signals from the same emitter together, provided additional context for the model during the reconstruction process that yielded the best performance. Further, the reconstruction-based transfer learning methods were more capable of overcoming differences in protocol between the source and target dataset.

*Universal representation learning*, a pre-training method, has also been explored in the context of RFML, but has yet to become as ubiquitous in the CV and NLP fields [38,40,41,42]. Universal representation learning approaches aim to learn general purpose features or embeddings that “capture the generic factors of variation present in all the classes” and that can be used between tasks [43]. Such approaches are then used as feature extractors or are fine-tuned for the target task(s). That is, universal representation learning is a source task that aims to provide successful transfer to a variety of target tasks, significantly decreasing training time for downstream algorithms.

## 5. Future Work

Every area of TL in the context of RFML remains an open area of research. Therefore, algorithmic development is the most apparent direction for future work. Most readily, parallels can be drawn between RFML and other modalities in which TL has been employed successfully in an effort to identify existing methods that can be borrowed. For example, many of the DL architectures used in RFML are CNN-based, such as in CV, yielding a large selection of TL methods from which to work from. Alternatively, the sequential nature of raw RF data is more akin to that used in text or speech-based language modeling, yielding additional NLP TL methods. However, borrowing such approaches yields no guarantee of success. Just as TL in text domains is characteristically different from TL in visual domains [44], thereby requiring different approaches and techniques, it is possible that wholly new TL algorithms will need to be developed for the RFML space.

Designing TL algorithms for the RFML space will first require a fundamental understanding of how both the channel environment and platform variations impact learned behavior and inhibit or facilitate transfer, which has yet to be thoroughly investigated. Even for RFML works that have successfully used TL, such limits in understanding may hinder further performance improvements that might be yielded from TL techniques. Finally, these fundamental limitations in the understanding also obscures insights into long-term model behavior during deployment, which has long been a criticism of RFML and prevented commercial support and deployment [14].

While existing works have addressed environment adaptation and environment platform co-adaptation via sequential learning techniques, little-to-no work has been performed to systematically examine the extent to which environmental factors and hardware considerations impact learned behavior and may, therefore, encourage or prevent transfer. One approach to performing such an analysis is to utilize transferability or distance metrics such as the LEEP score [45], LogME score [46], or Proxy-A distance [47] to quantify the distance between datasets and/or trained NNs. Such metrics can be examined as a function of domain and task metadata parameters-of-interest using sensitivity analysis.

Similarly, while existing works have utilized multitask and sequential learning techniques with success, no work has examined the changes in learned behavior between models trained using TL techniques versus random initialization. Methods for understanding pre-trained NNs, often through the use of visualization techniques or mathematical theory [48,49,50,51,52], are still a very active area of research [53]. Some recent works have used Canonical Correlation Analysis to compare NN representations, and they have been used to examine how learned representations change throughout training and differ when trained from different random initializations [54,55,56]. Such an approach could be also used for studying changes in learned behavior between models trained from random initialization, those that have been fine-tuned using sequential learning methods, and those trained using multitask learning techniques.

Finally, as the field of RFML grows to encompass unsupervised and self-supervised techniques, this taxonomy will need to be expanded to include transductive learning techniques such as those presented in the general taxonomy of [9].

## 6. Conclusions

TL has become a transformative technology in the CV and NLP modalities for improving the performance of DL algorithms with less training data by utilizing knowledge gleaned from similar domains and/or tasks. In light of recent research highlighting the necessity for and extraordinary cost of representative captured RF training data for RFML [8], TL is a promising future capability for RFML, but it has been underutilized thus far. Furthermore, TL is a requisite technology for realizing online and distributed learning techniques for RFML, where the propagation environment and RF hardware have a direct impact on received data and, therefore, the performance of the resultant algorithm. This paper has presented a TL taxonomy for RFML and has surveyed the small body of existing works utilizing TL in the context of RFML, including the methods used and patterns observed in the results across these existing works. In performing this, this work has offered a framework through which broader research efforts in TL for RFML can be categorized, compared, and contrasted, and it has highlighted areas where future work is needed in order to mature the field.

## Figures and Tables

**Figure 1 sensors-22-01416-f001:**
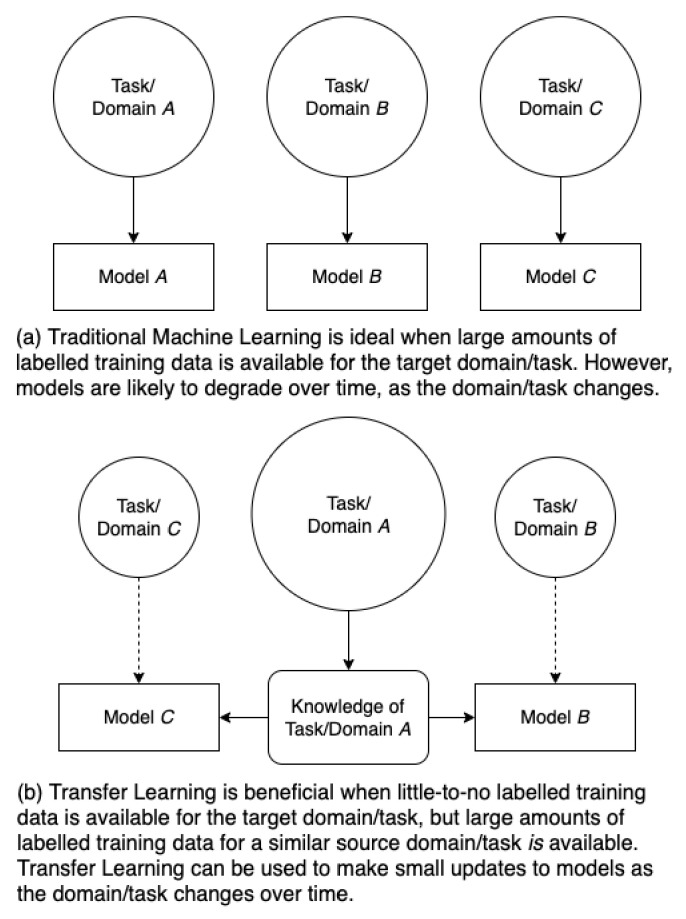
The difference between traditional ML (**a**), in which a new model is trained on a each domain/task pairing from random initialization, and TL (**b**), in which prior knowledge learned on one domain/task is used to support performance on a second domain and/or task where less (or no) labelled data were available.

**Figure 2 sensors-22-01416-f002:**
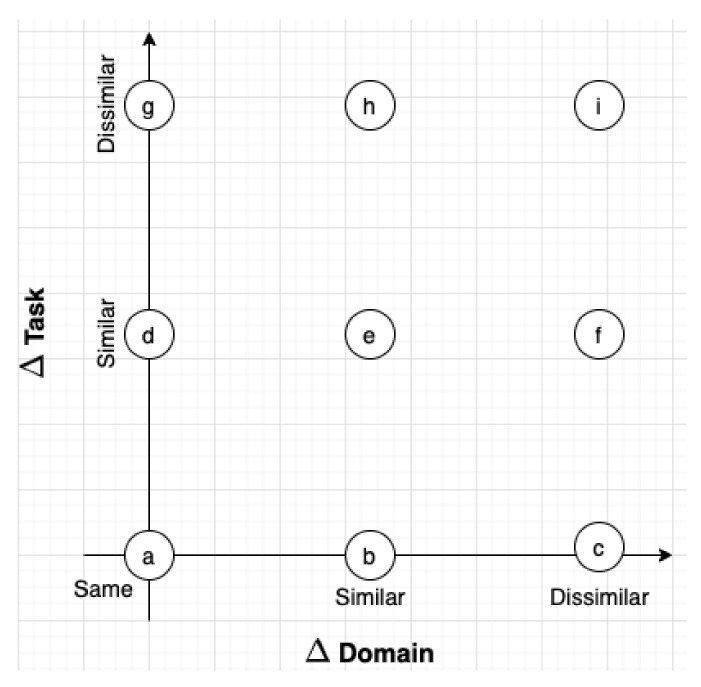
The two-dimensional spectrum of “similarity” between source and target domains and tasks, with the origin (**a**) representing the same task and domain. For clarity, the settings that describe (**a**–**i**) are provided in Table 2.

**Figure 3 sensors-22-01416-f003:**
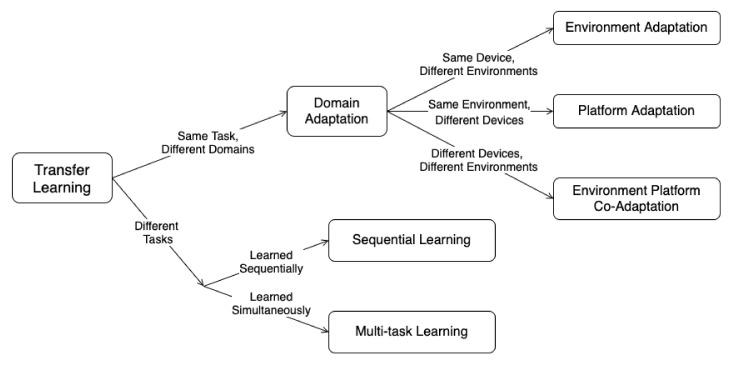
The proposed TL taxonomy for RFML.

**Table 1 sensors-22-01416-t001:** Example RFML domain elements and tasks.

Domain Elements	Tasks
SNR AWGN Ricean Fading Multipath Effects Doppler Bandwidth Sample Rate Noise Floor IQ Imbalance Phase Imbalance Non-linear distortion	*n*-class AMC SEI Localization Signal Detection End-to-End Communications SNR Estimation IQ Imbalance Estimation Signal Compression

**Table 2 sensors-22-01416-t002:** The settings that describe points (a–i) on the two-dimensional spectrum of “similarity” between source and target domains and tasks shown in Figure 2.

Setting	Description
(a)	The traditional ML setting where the source and target domains and tasks are the same.
(b)	The TL setting in which learned features from one domain are used to support performing the same task in a second domain. For example, using features learned to perform AMC in an AWGN channel to support performing AMC in a fading channel.
(c)	The setting in which source and target domains are so dissimilar that TL is unsuccessful, despite the source and target tasks being the same.
(d)	The TL setting in which learned features from one task are used to support a second task, while the source and target domains are the same. For example, using features learned to perform AMC to support SEI with the source and target domains being the same.
(e)	Likely the most challenging TL setting in which learned features from one domain and task are used to support performing a second task in a new domain. For example, using features learned to perform AMC in an AWGN channel to support performing SEI in a fading channel.
(f)	The setting in which source and target domains are so dissimilar that TL is unsuccessful, although the source and target tasks are somewhat similar.
(g)	The setting in which source and target tasks are so dissimilar that TL is unsuccessful, despite the source and target domains being the same.
(h)	The setting in which source and target tasks are so dissimilar that TL is unsuccessful, despite the source and target domains being somewhat similar.
(i)	The setting in which both source and target tasks and domains are dissimilar, preventing the use of successful TL.

**Table 3 sensors-22-01416-t003:** Representative examples for TL settings in RFML.

TL Setting	Use Case	Source Domain	Source Task	Target Domain	Target Task
Environment Adaptation	Move a Tx/Rx pair equipped with an AMC model from an empty field to a city center	Single Tx/Rx pair, AWGN channel	Binary AMC (BPSK/QPSK)	Same Tx/Rx pair, Multipath channel	Binary AMC (BPSK/QPSK)
Platform Adaptation	Transfer an AMC model between UAV	Single Rx, Many Tx, Fading channel w/ Doppler	Binary AMC (BPSK/QPSK)	Different Rx, Same Tx set, Fading channel w/ Doppler	Binary AMC (BPSK/QPSK)
Environment Platform Co-Adaptation	Transfer an AMC model between a ground-station and UAV	Single Rx, Many Tx, Multipath channel	Binary AMC (BPSK/QPSK)	Different Rx, Same Tx set, Fading channel w/ Doppler	Binary AMC (BPSK/QPSK)
Multitask Learning	Simultaneous signal detection and AMC	Single Tx/Rx pair, AWGN channel	Binary AMC (BPSK/QPSK)	Same Tx/Rx pair, AWGN channel	SNR Estimation
Sequential Learning	Addition of an output class(es) to an	Single Tx/Rx pair, AWGN channel	Binary AMC (BPSK/QPSK)	Same Tx/Rx pair, AWGN channel	Four-class AMC (BPSK/QPSK/ 16QAM/64QAM)

## Data Availability

Not applicable.

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
