# Peer review of "Transfer Learning for Radio Frequency Machine Learning: A Taxonomy and Survey"

_sensors, 2022, doi:10.3390/s22041416_

Round 1

Reviewer 1 Report

In this work, a comprehensive summary of works related to transfer learning for radio frequency machine learning is made. The authors also propose a novel framework for describing the relationship between the works. This paper is clearly written and well organized. The reviewer thinks this paper can be accepted for publication. 

Various deep learning techniques including transfer learning play a critical role in radio frequency communication in recent years. In this work, a comprehensive summary of previous works related to transfer learning (TL) for radio frequency machine learning (RFML) is made. The authors also propose a novel framework for describing the relationship between these previous works. As far as the author knows, this is the first literature review specially focused on TL for RFML. The work has certain novelty and originality. This paper is clearly written and well organized. The references are adequate and appropriate. The reviewer thinks this paper can be accepted for publication.

Author Response

Dear Reviewer,

Thank you for your time and consideration in reviewing our manuscript.

Best,

Lauren Wong

Reviewer 2 Report

Dear Authors,

Thank you very much for submitting the article entitled: "Transfer Learning for RFML: A Taxonomy and Survey" to Sensors.

The first doubt I have is related to the journal, in the manuscript it is stated that the publication has been sent to Signals.

Related to the title, I think it is not interesting to introduce the acronym in the title, I think it is better to incorporate it in the keywords.

The article uses many acronyms, acronyms should be described the first time they are used in the main text of the manuscript. It is generally recommended not to use acronyms in the abstract.

I have a question about the procedure that has been used to develop the taxonomy, what methodology have the researchers used to develop the taxonomy, what is the procedure for selecting the papers under study?

I hope that my comments and suggestions will be useful in future versions of the paper.

Author Response

Dear Reviewer,

Thank you for your time and consideration in reviewing our manuscript.

Regarding your first concern, I had previously been mistaken in thinking that the special issue "Radio Frequency Machine Learning Applications" was a special issue belonging to the journal "Signals." This was my error. Regardless I believe this special issue to be the best venue for the work.

As requested, I have ensured that all acronyms are described in their first use, and have fully spelled out Radio Frequency Machine Learning in the title.

To address your question regarding the procedure: This taxonomy was built off of a generalized transfer learning taxonomy accepted within the machine learning community (Pan and Yang 2010). It was then customized to suit the unique challenges and needs of RFML use cases, such as platform and environment adaptation. The RF transfer learning taxonomy constructed was refined to ensure all RFML works using transfer learning techniques, identified during a thorough literature review, are represented.

Best,

Lauren Wong

Reviewer 3 Report

Title: Transfer Learning for RFML: A Taxonomy and Survey

Authors: Lauren J. Wong, and Alan J. Michaels

Recommendation: Publish after minor revision.

This manuscript by Wong et al. presents a tailored transfer learning (TL) taxonomy for radio frequency applications to generate a framework that can be used to compare and analyze among existing and other works. It is to the credit of the authors to highlight the promising capability of a TL taxonomy for RFML, which requires less training data and utilizes information collected from similar tasks. Due to the high quality of the investigation and the care with which this manuscript was prepared there is little ground for review. Nonetheless, the authors are encouraged to make corrections addressing below points before publication.

- In current Figure 1, it is not immediately clear the difference and comparison between TL versus ML and DL. A table clearly stating the pros and cons for each will greatly help readers’ information.

- Missing physical address for the “Intel AI Lab” affiliation.

- On Page 1, Line 15, the authors should spell “ML”, “DL”, and “RFML” for the first-time use.

- On Page 4, Line 112, change “For example, …” to “On the one hand, …”.

- On Page 7, Line 191, change to “i.e.,”.

Author Response

Dear Reviewer,

Thank you for your time and consideration in reviewing our manuscript.

As requested, I have ensured that all acronyms are described in their first use, and added a physical address for Intel AI Lab. I have also made the recommended changes on page 4, line 112 and page 7, line 191.

With regards to figure 1, I have included more detailed figure descriptions to help the reader understand the pros/cons of traditional machine learning and transfer learning techniques.

Best,

Lauren Wong

Round 2

Reviewer 2 Report

Dear authors,

Thank you very much for considering my comments and suggestions.